# Daytime-only-mean data enhances understanding of land-atmosphere coupling

Zun Yin[1], Kirsten L. Findell[2], Paul Dirmeyer[3], Elena Shevliakova[2], Sergey Malyshev[2], Khaled Ghannam[1], Nina Raoult[4], and Zhihong Tan[1]

[1]Program in Atmospheric and Oceanic Sciences, Princeton University, Princeton, 08540, New Jersey, USA
[2]Geophysical Fluid Dynamics Laboratory, NOAA, Princeton, 08540, New Jersey, USA
[3]Center for Ocean-Land-Atmosphere Studies, George Mason University, Fairfax, 22030, Virginia, USA
[4]Laboratoire des Sciences du Climat et de l'Environnement, IPSL, CNRS-CEA-UVSQ, Gif-sur-Yvette, 91191, Essonne, France

**Correspondence:** Zun Yin (zyin@princeton.edu)

**Abstract.** Land-atmosphere (L-A) interactions encompass the co-evolution of the land surface and overlying planetary boundary layer, primarily during daylight hours. However, many studies have been conducted using monthly or entire-day-mean time series due to the lack of sub-daily data. It is unclear whether the inclusion of nighttime data alters the assessment of L-A coupling or obscures L-A interactive processes. To address this question, we generate monthly (M), entire-day-mean (E), and daytime-only-mean (D) data based on the ERA5 (5th European Centre for Medium-Range Weather Forecasts reanalysis) product, and evaluate the strength of L-A coupling through two-legged metrics, which partition the impact of the land states on surface fluxes (the land leg) from the impact of surface fluxes on the atmospheric states (the atmospheric leg). Here we show that the spatial patterns of strong L-A coupling regions among the M-, D- and E-based diagnoses can differ by more than 80%. The signal loss from E- to M-based diagnoses is determined by the memory of local L-A states. The differences between E- and D-based diagnoses can be driven by physical mechanisms or averaging algorithms. To improve understanding of L-A interactions, we call attention to the urgent need for more high-frequency data from both simulations and observations for relevant diagnoses. Regarding model outputs, two approaches are proposed to resolve the storage dilemma for high-frequency data: (1) integration of L-A metrics within Earth System Models, and (2) producing alternative daily datasets based on different averaging algorithms.

## 1 Introduction

Numerous studies have demonstrated the importance of land-atmosphere (L-A) interactions to the earth system (Findell et al., 2011; Hu et al., 2021; Klein and Taylor, 2020; Laguë et al., 2019; Taylor et al., 2012). Manifested by the mass and energy exchanges between the land surface and the planetary boundary layer (PBL), L-A interactions influence the evolution of convective systems (Hu et al., 2021; Klein and Taylor, 2020) as well as the occurrence of convective rainfall (Taylor et al., 2012). From a climatic perspective, coupling processes between the land and the atmosphere can accelerate the frequency and intensity of extreme events (Dirmeyer et al., 2021; Miralles et al., 2019; Schumacher et al., 2019; Zhou et al., 2021) and the shift

of climate regimes (Berg et al., 2017; Findell et al., 2019) under global warming. To better understand L-A interactions, a suite of metrics has been proposed for characterizing specific physical processes across broad spatial and temporal scales (Santanello et al., 2018). These metrics can reveal essential behaviors of L-A interactions and enhance our understanding of the coupling mechanisms (e.g., Chen and Dirmeyer (2017); Findell et al. (2011); Hu et al. (2021); Jach et al. (2022)). Additionally, they provide a benchmark to evaluate the performance of earth system models in simulating L-A coupling processes (e.g., Dirmeyer et al. (2018); Ferguson et al. (2012); Koster et al. (2006); Santanello et al. (2009)).

However, L-A interactions alone are not always the primary determinant in the climate system (Koster et al., 2004). To reveal hotspots where and when L-A interactions play an important role, two criteria have been proposed: 1) the state of the atmosphere must be highly responsive to variations in land properties, and 2) there must be physically meaningful variability in those land properties over time (Dirmeyer, 2011; Guo et al., 2006; Koster et al., 2004). Dirmeyer (2011) proposed a metric ($M$) to characterize both features as

$$M = \frac{\mathrm{d}b}{\mathrm{d}a} \cdot \sigma_a = \rho(a,b)\sigma_b. \tag{1}$$

$M$ contains two components to estimate the coupling strength between variables $a$, presumed to be the driver, and $b$, the response. The coupling is significant only when $b$ is sensitive to $a$ (high $\mathrm{d}b/\mathrm{d}a$) and the variation of $a$ (standard deviation of $a$, $\sigma_a$) is large. The formula is equivalent to the correlation coefficient between $a$ and $b$ (i.e., $\rho(a,b)$) multiplied by $\sigma_b$. The advantage of this metric is its vast suitability in characterizing coupling mechanisms across different scales (Chen and Dirmeyer, 2017; Guillod et al., 2014; Hu et al., 2021; Lorenz et al., 2015) regardless of specific variables. In terms of L-A interactions, Dirmeyer et al. (2014) divided the coupling linkage into two steps: a land leg capturing the coupling between the land surface state (typically characterized by soil moisture) and surface fluxes of heat, moisture, or momentum, and an atmospheric leg capturing the coupling between the surface fluxes and the atmosphere states (see Sect. 2.2).

The two-legged metrics (TLMs) mainly focus on processes operating in response to daytime solar heating. However, data covering daylight hours is rare in available datasets. Consequently, most TLM research has been based on time series of monthly or 24-hour average quantities (e.g., Dirmeyer et al. (2014); Hu et al. (2021); Lorenz et al. (2015)). Although these studies enhance our understanding of the patterns and seasonality of L-A coupling, little has been done to show whether the monthly- and entire-day-based inputs are able to accurately capture areas with strong daytime land-atmosphere coupling (Seo and Dirmeyer, 2022). In other words, are there significant differences among monthly-, entire-day-, and daytime-only-based L-A coupling diagnoses? If so, are the differences exclusively due to the averaging process, or are there other L-A coupling mechanisms that may mislead the diagnoses of daytime L-A coupling?

In this study, the $0.25°$ spatial resolution ERA5 (the fifth ECMWF reanalysis, (Hersbach et al., 2018)) is employed as the test bed to address these research questions. Three time series derived from ERA5 outputs, monthly-means (M), entire-day-means (E), and daytime-only-means (D), are utilized to calculate two-legged metrics (TLMs) to evaluate L-A coupling strength. We investigate the spatial pattern differences among M-, E-, and D-based diagnoses. Primary contributors to the pattern mismatch are revealed, associated mechanisms are demonstrated, and implications are discussed.

## 2 Methods

### 2.1 ERA5 data

The ERA5 reanalysis provides $0.25°$-hourly modeling estimates assimilated with historical observations (e.g., soil moisture, 10-m wind, 2-m humidity, and temperature (Hersbach et al., 2020)). We collected ERA5 output over land (land-ice included) every other hour from 1:00 UTC (Coordinated Universal Time) 01-Jan-2011 until 23:00 UTC 31-Dec-2020 over $[180°\text{W}$–$180°\text{E}]\times[65°\text{S}$–$80°\text{N}]$. To be consistent with other daily datasets, the entire-day-mean values (E) are obtained by averaging time steps within each day based on the UTC. For the daytime-only-mean (D), the globe is divided into twenty-four time zones and the time is converted from UTC to LST (Local Solar Time). The time steps between 8am and 6pm LST are averaged to generate D values. The monthly mean (M) is a monthly average of E. To meet the minimum length requirement (Findell et al., 2015) for monthly TLMs estimations, we collected forty years of M data from 1981 through 2020.

There are multiple ways of describing the linkages between the land, surface fluxes, and the atmosphere that the TLM are meant to capture. For instance, the land leg can be structured to investigate how the land affects convective precipitation via the latent heat flux, or how the land influences the growth of the planetary boundary layer (PBL) through the sensible heat flux. As it is difficult to distinguish L-A triggered convective precipitation, we select the latter in this study, using surface soil moisture from the 0–7 cm soil layer ($\theta$ [m$^3$.m$^{-3}$]) and sensible heat flux ($H$ [W.m$^{-2}$]) to characterize the land leg. Additionally, to enable validation of ERA5 data with ground-based observations (i.e., FLUXNET, validation results are not shown) that lack observed PBL heights, we select the pressure at the lifting condensation level ($P_{\text{lcl}}$ [Pa]) to represent the atmospheric state, specifically that of the PBL. $P_{\text{lcl}}$ can be estimated from three regular ground measurements: the surface pressure ($P$ [Pa]), 2-m temperature ($T_{2\text{m}}$ [K]), and 2-m dew-point temperature ($D_{2\text{m}}$ [K]) (Georgakakos and Bras, 1984), as:

$$P_{\text{lcl}} = P - P\left(\frac{T_{2\text{m}} - D_{2\text{m}}}{223.15} + 1\right)^{-3.5}. \tag{2}$$

The three time series are grouped by season. Both long-term trends and seasonality are removed to prevent them from obscuring the signal and altering the diagnoses, following Dirmeyer et al. (2012).

### 2.2 Two-legged metrics

The two-legged metrics (TLMs) contain a land leg and an atmospheric leg to evaluate the two coupling links in the L-A interaction chain (Dirmeyer et al., 2014; Santanello et al., 2018). If $\theta$, $H$, and $P_{\text{lcl}}$ are utilized to represent the states of the land, the surface flux, and the atmosphere, the L-A coupling metrics (Eq. 1) can be formulated to assess the two-stepped coupling processes as:

$$
\begin{aligned}
\mathcal{L} &= \frac{\mathrm{d}H}{\mathrm{d}\theta}\sigma_\theta = \rho(\theta, H)\cdot\sigma_H, \\
\mathcal{A} &= \frac{\mathrm{d}P_{\text{lcl}}}{\mathrm{d}H}\sigma_H = \rho(H, P_{\text{lcl}})\cdot\sigma_{P_{\text{lcl}}}, \\
\mathcal{T} &= \frac{\mathrm{d}H}{\mathrm{d}\theta}\frac{\mathrm{d}P_{\text{lcl}}}{\mathrm{d}H}\sigma_\theta = \rho(\theta, H)\rho(H, P_{\text{lcl}})\cdot\sigma_{P_{\text{lcl}}}.
\end{aligned}
\tag{3}
$$

 $\mathcal{L}$, $\mathcal{A}$, and $\mathcal{T}$ indicate the land, the atmospheric, and the total legs, respectively. By applying Eq. 3 to the M, E, and D time series, we get different versions of TLMs, denoted by $\text{TLM}_\text{M}$, $\text{TLM}_\text{E}$, and $\text{TLM}_\text{D}$, respectively. For a specific variable and leg, we use M, E, and D as subscripts to distinguish them (e.g., $\mathcal{L}_\text{M}$, $\mathcal{L}_\text{E}$, and $\mathcal{L}_\text{D}$).

## 2.3 Spatial pattern comparisons among M-, E-, and D-based diagnoses

The TLMs are designed to highlight differences in L-A coupling strength between geographic regions and/or between different times of year in a given region. Those relative differences require subjective decisions to determine the threshold values separating regions of "strong" coupling from regions of weaker coupling. However, a direct comparison of the numerical values of TLMs based on different time windows of inputs (i.e., M, E, and D) is not appropriate for three primary reasons. First, the magnitude of the TLMs is strongly affected by the $\sigma$ term (Eq. 1), and this measure of variability can be quite different for daytime and nighttime processes. For example, D-based $H$ and $P_\text{lcl}$ have much larger variances than that based on the entire-day-mean, which systematically enlarges the $\mathcal{L}_\text{D}$ and $\mathcal{A}_\text{D}$. Additionally, strong L-A coupling signals can be positive or negative, suggesting that the change of TLM's magnitude (its absolute value) is the relevant quantity of interest rather than the magnitude of changes. Finally, L-A coupling processes are not characterized by clear thresholds, but rather by relative spatial and temporal differences.

To overcome these limitations and remove any subjectivity in our assessment of coupling strength, we use quantiles to assess coupling strengths and quantify the spatial differences between $\text{TLM}_\text{M}$, $\text{TLM}_\text{E}$, and $\text{TLM}_\text{D}$. The quantile approach can reflect the spatial patterns of TLM and provide the possibility of pattern comparison between TLMs based on different inputs. Other climate-relevant studies have also successfully utilized the quantile approach to compare estimates based on different algorithms. For example, because satellite-based and modeled estimations are not suitable for direct comparison with gauge measurements, the quantile approach was employed for relevant bias correction or downscaling in the form of probability density functions (PDF) Guo et al. (2018); Vrac et al. (2012); Xie et al. (2017). For a specific TLM and a given quantile threshold, regions with absolute values of TLM over this threshold are marked for each of the M, D, and E cases. For the $\mathcal{A}_\text{D}$ in a specific period for example, if the given threshold is 0.8, grid cells with the top 20% largest $|\mathcal{A}|$ are marked. The ratio of the number of overlapping grid cells to the number of E-based marked grid cells is defined as the fitting rate between $\mathcal{A}_\text{E}$ and $\mathcal{A}_\text{D}$, which can reflect the difference between D- and E-based diagnoses at different levels of coupling strength. The same approach is applied to the legs in paired comparisons of E vs M, M vs D, and D vs E.

## 2.4 Signal attenuation from $\text{TLM}_\text{E}$ to $\text{TLM}_\text{M}$

The TLMs contain a correlation term $\rho$ and a variance term $\sigma$ (Eq. 1). First, we investigate the difference of the $\sigma$ term between E- and M-based TLMs. To keep the symbols simple, we denote $a_i$ and $b_i$ ($i$ is day index) as the detrended and seasonality-removed daily time series. $A_j$ and $B_j$ ($j$ is the month index) are corresponding monthly time series. As the long-term average

of $b_i$ (i.e., $\bar{b}$) is zero, the $\sigma_b$ can be expressed as

$$\sigma_b = \left( \frac{1}{DMY} \sum_{i=1}^{DMY} b_i^2 - \bar{b}^2 \right)^{\frac{1}{2}}$$

$$= \left( \frac{1}{MY} \sum_{i=1}^{MY} \left[ \frac{b_i^2 + b_{i+1}^2 + b_{i+2}^2 + ... + b_{i+D}^2}{D} \right]_j \right)^{\frac{1}{2}} \tag{4}$$

$D$, $M$, and $Y$ are the number of days, months, and years, respectively. The $\sigma_B$ can be written as

$$\sigma_{B_j} = \left( \frac{1}{MY} \sum_{j=1}^{MY} B_j^2 - \bar{B}^2 \right)^{\frac{1}{2}}$$

$$= \left( \frac{1}{MY} \sum_{j=1}^{MY} \left( \frac{\sum_{i \in j}^{D} b_i}{D} \right)^2 \right)^{\frac{1}{2}}$$

$$= \left( \frac{1}{MY} \sum_{j=1}^{MY} \left[ \frac{(b_i + b_{i+1} + b_{i+2} + ... + b_{i+D})^2}{D^2} \right]_j \right)^{\frac{1}{2}} \tag{5}$$

$\sigma_b$ contains all squared $b_i$, but $\sigma_B$ contains averaged products of all combinations of $b_i$ within a month. It is not difficult to prove that $D^2 \sum_{i=1}^{N} b_i^2 \geq (b_i + b_{i+1} + ... + b_N)^2$. The equal relation stands when $b_i = b_{i+1} = ... = b_N$, indicating all daily variables are the same within a month. Considering all months, the $\sigma_B$ is larger if $b_i$ follows the Matthew principle better, that is large values assemble together in specific months and small values assemble together in other months. As $b_i$ is a time series of variables in a natural process. $b_i$ is somehow correlated with itself at a certain time scale, that is the memory of $b_i$. It implies that if $b_i$ is large, its neighbours (e.g., $b_{i-1}$ and $b_{i+1}$) are large as well. Thus, the memory (characterized by auto-correlation) may determine the information maintained from $\sigma_b$ to $\sigma_B$, if the $\sigma_b$ is considered as the accurate information we want.

The $\rho$ term based on daily time series can be written as:

$$\rho(a,b) = \frac{\sum_{i=1}^{DMY} (a_i - \bar{a})(b_i - \bar{b})}{\sigma_a \sigma_b}$$

$$= \frac{\sum_{i=1}^{DMY} a_i b_i}{\sigma_a \sigma_b}. \tag{6}$$

$\bar{a}$ and $\bar{b}$ are mean of $a_i$ and $b_i$, respectively. Similarly, we can get $\rho(A,B)$ as

$$\rho(A,B) = \frac{\sum_{j=1}^{MY} (A_j - \bar{A})(B_j - \bar{B})}{\sigma_A \cdot \sigma_B}$$

$$= \frac{1}{\sigma_A \sigma_B} \sum_{j=1}^{MY} \left( \frac{\left( \sum_{i \in j} a_i \right)\left( \sum_{i \in j} b_i \right)}{D^2} \right). \tag{7}$$

The $\rho$ terms contain $\sigma$ terms, which have been discussed. If we focus on the numerator, we can find that the difference of numerator between E and M has a similar structure as the $\rho$ difference between E and M. Thus, we deduce that the cross-covariance between $a_i$ and $b_i$ is the key contributor to the difference of the $\rho$'s numerator between E and M.

According to our deduction, we infer that the memory of the L-A state (i.e., the auto-correlation for a single variable and the cross-covariance for paired variables) can characterize the coupling signal attenuation due to the monthly smoothing of daily time series. Thus, for a single variable (i.e., the $\sigma$ term), we calculate its auto-correlation function (ACF) with a maximum lag of 30 days (within a month). Then we average the ACF values belonging to the top 25% quantile as an indicator of the attenuation resistance (Supplementary Fig. S1a). The attenuation resistance is characterized by the ratio of $\sigma_M$ to $\sigma_E$. For paired variables (i.e., the numerator of the $\rho$ term $N(\rho)$, e.g., $N(\rho) = \sum_{i=1}^{DMY} a_i b_i$ in Eq. 6), we calculate the cross-covariance function (CCF) instead, but with a maximum lag $\pm 30$ days. For negatively correlated variables, we select the mean of the lowest 25% CCF as the indicator (Supplementary Fig. S1b). For positively correlated variables, we select top 25% as the quantile threshold as the ACF case (Supplementary Fig. S1c). Instead of $N(\rho_M)/N(\rho_E)$, we use $N(\rho_M)/(|N(\rho_E)|+|N(\rho_M)|)$ to characterize associated signal attenuation resistance, in order to avoid uncertainties due to phase shift from $N(\rho_E)$ to $N(\rho_M)$.

## 2.5 $\Delta|\text{TLM}|$ decomposition

According to the form of the coupling metrics (Eq. 1), the differences among $|\text{TLM}_M|$, $|\text{TLM}_E|$, and $|\text{TLM}_D|$ can be decomposed using $M_1$ and $M_2$ as specific TLMs based on two different time series:

$$
\begin{aligned}
\Delta|M| &= |M_2| - |M_1| \\
&= C_\rho + C_\sigma + C_{\sigma\rho}, \text{where} \\
C_\rho &= \sigma_1 \left( |\rho_2| - |\rho_1| \right) \\
C_\sigma &= |\rho_1| \left( \sigma_2 - \sigma_1 \right) \\
C_{\sigma\rho} &= \left( |\rho_2| - |\rho_1| \right) \left( \sigma_2 - \sigma_1 \right).
\end{aligned}
\tag{8}
$$

$\Delta|M|$ is the absolute value (coupling strength) shift from $M_1$ to $M_2$, which is composed of contributions from the correlation term ($C_\rho$), the fluctuation term ($C_\sigma$), and the joint term ($C_{\sigma\rho}$). Note that the three contributing terms may be either positive or negative. Thus, we take their absolute values to estimate their fractional contributions to the total coupling strength shift, $\Delta|M|$. For example, the fractional contribution of the correlation term is calculated as:

$$
\frac{|C_\rho|}{|C_\rho| + |C_\sigma| + |C_{\sigma\rho}|}.
\tag{9}
$$

### 2.6 Primary contributors to TLM pattern shift

As discussed in Section 2.3, describing TLMs with quantiles brings a focus to spatial patterns and regions of strong coupling, relative to neighboring regions. This approach can be extended to describe the shifts in spatial patterns from $M_1$ to $M_2$ using quantile changes ($\Delta q$). This is a better descriptor of changes in spatial patterns than $\Delta|\text{TLM}|$, because the latter only quantifies the value changes within a specific grid cell, which cannot reflect the relative TLM change among grid cells. Moreover, within $C_\rho$, $C_\sigma$, and $C_{\sigma\rho}$, the largest contributor (Eq. 8 and 9) to $\Delta|\text{TLM}|$ may not be the dominant factor for $\Delta q$ of specific grid cells. For example, one grid cell has an increase from $|M_1|$ to $|M_2|$ with $[C_\rho = 0, C_\sigma = 100, C_{\sigma\rho} = 20]$, but another grid cell has an increase with $[C_\rho = 0, C_\sigma = 100, C_{\sigma\rho} = 0]$. The first grid cell has a non-zero $\Delta q$, but the component that determines the $q$

increase is not the largest contributor to $\Delta|M|$ (i.e., $C_\sigma$), but rather the $C_{\sigma\rho}$. The dominant factor of a specific grid cell must be the one without which the quantile of the grid cell has the lowest change from $\text{TLM}_1$ to $\text{TLM}_2$.

To demonstrate the dominant factor leading to $\Delta q$ for a specific grid cell, we calculate $\Delta q$ in four scenarios:

$$\Delta q = q_{|M_2|} - q_{|M_1|}$$

$$\Delta q_{\rho-} = q_{|M_2|-C_\rho} - q_{|M_1|}$$

$$\Delta q_{\sigma-} = q_{|M_2|-C_\sigma} - q_{|M_1|}$$

$$\Delta q_{\sigma\rho-} = q_{|M_2|-C_{\sigma\rho}} - q_{|M_1|}. \tag{10}$$

$\Delta q$ is the $q$ shift of a specific grid cell from $|M_1|$ to $|M_2|$. $\Delta q_{\rho-}$ is the $q$ shift without the contribution of the $\rho$ term (i.e., from $|M_1|$ to $|M_2| - C_\rho$). Similar definitions are applied for $\Delta q_{\sigma-}$ and $\Delta q_{\sigma\rho-}$. Then we can demonstrate the dominant factor for a specific grid cell as:

$$f_{\min}\left(\Delta q_{\rho-}, \Delta q_{\sigma-}, \Delta q_{\sigma\rho-}\right), \text{ if } \Delta q > 0,$$

$$f_{\max}\left(\Delta q_{\rho-}, \Delta q_{\sigma-}, \Delta q_{\sigma\rho-}\right), \text{ if } \Delta q < 0. \tag{11}$$

$f_{\min}$ ($f_{\max}$) is a function selecting the corresponding subscript of the term with the minimum (maximum) value.

## 3 Results

### 3.1 Spatial pattern differences among diagnoses based on $\text{TLM}_\text{M}$, $\text{TLM}_\text{E}$, and $\text{TLM}_\text{D}$

Using ERA5 hourly data, we generated three homologous time series with three different temporal averaging algorithms: monthly mean (M), entire-day-mean (E), and daytime-mean (D). These three time series were used to estimate the coupling strength between the land and the atmosphere based on the two-legged metrics (Eq. 3, Sect. 2.2). Figure 1 assesses the geographic consistency between the coupling strengths determined by the three different time series by showing the fitting rate of a suite of comparisons at different levels of quantile thresholds (Sect. 2.3). In all seasons, $\mathcal{A}$ has a much lower fitting rate than $\mathcal{L}$, and the fitting rate of $\mathcal{T}$ lies between the two. This is a reflection of the long memory inherent in the land relative to the atmosphere. In addition, fitting rates vary with season, and JJA has the lowest value, indicating that the largest spatial difference occurs in the summer of the Northern Hemisphere where most land is located. The median of fitting rates over all legs and seasons is 69.4% if the largest 10% of TLM values are considered physically significant, demonstrating that the determination of L-A coupling strongly depends on the averaging time period of the input time series. Most fitting rates decrease with the rise of the quantile threshold, and the lowest fitting rate is 15.2% ($\mathcal{A}_\text{M}$ vs. $\mathcal{A}_\text{D}$ in JJA for the 0.95 quantile threshold), indicating that only a small portion of the most strongly coupled regions (the top 5%) are simultaneously diagnosed by both D and M. To focus on the season and coupling leg with the largest sensitivity to time series averaging window, we select $\mathcal{A}$ in summer (JJA and DJF in the Northern and Southern Hemisphere, respectively) as an example to explore the TLM differences in the following content.

Figure 2a illustrates the differences of strong L-A coupling regions (90% quantile as the threshold) among $\mathcal{A}_M$, $\mathcal{A}_D$, and $\mathcal{A}_E$ during each hemisphere's summer season. Although the total area of overlap ($\mathcal{A}_M \cap \mathcal{A}_E \cap \mathcal{A}_D$, pale taupe area in Fig. 2a) accounts for approximately 50% of strong coupling regions, vast disagreement among those diagnoses still exist, especially in the Northern Hemisphere. $\mathcal{A}_M$ suggests strong coupling in some climate transition regions (such as the western and southern US, central Asia, northern India, eastern Sahel, and southern Australia). $\mathcal{A}_E$ highlights some mid-latitude regions, such as the southwestern US, a part of the Sahara, Arabia, central India, and northwestern China. However, as the most accurate diagnosis, $\mathcal{A}_D$ demonstrates that the L-A coupling is stronger in the southeastern US and in high latitudes, such as the boreal forest region of Canada, and parts of northern Eurasia. Interestingly, the fraction of $\mathcal{A}_M \cap \mathcal{A}_D$ (1.7%) is much less than that of $\mathcal{A}_M \cap \mathcal{A}_E$ (7.6%) or $\mathcal{A}_E \cap \mathcal{A}_D$ (11.5%), implying that $\mathcal{A}_E$ is the intermediate status between $\mathcal{A}_M$ and $\mathcal{A}_D$. Therefore, we investigate the two-stepped transitions: $\mathcal{A}_M \rightarrow \mathcal{A}_E$ (M vs E) and $\mathcal{A}_E \rightarrow \mathcal{A}_D$ (E vs D) in the following analysis.

Figure 2b shows the quantile transition of $\mathcal{A}_M \rightarrow \mathcal{A}_E$ in summer. Two types of regions are important. One is the green/yellow regions showing quantile shifts within the strongest coupling group, which coincide with the regions highlighted by Fig. 2a. The other is the dark blue/red regions, indicating the largest quantile changes from $\mathcal{A}_M$ to $\mathcal{A}_E$. Interestingly, the quantile drops dramatically in the center of North America, the Sahel, and central Asia. On one hand, those $\mathcal{A}_M$ diagnosed strongly L-A coupled regions agree with the findings from Koster et al. (2004) that was based on six-day averaged data. On the other hand, the coupling strength of those regions fades significantly when E-based diagnoses are applied. For instance, the quantile for three selected sites in these areas (red triangles in Fig. 2b) drops from $> 80\%$ ($\mathcal{A}_M$) to $< 30\%$ ($\mathcal{A}_E$). It indicates that the L-A coupling strength may be overestimated in those climatic transition zones if multi-day average data was applied. In the next section, we will demonstrate the mechanism resulting in such vast differences between $\mathcal{A}_M$ and $\mathcal{A}_E$.

Figure 2c displays the quantile transition of $\mathcal{A}_E \rightarrow \mathcal{A}_D$ in summer. In general, the most significant quantile shifts occur in the Northern Hemisphere and the strongly coupled regions are diagnosed further north by $\mathcal{A}_D$. The Sahara and Arabia contribute the largest quantile drop of $\mathcal{A}_E \rightarrow \mathcal{A}_D$. Some regions show strong coupling based on both $\mathcal{A}_E$ and $\mathcal{A}_D$. However, their coupling strength is overestimated by $\mathcal{A}_E$, such as the southwestern US and northern Mexico, India, and northwestern China. Key regions with increasing $\mathcal{A}$ quantile include the eastern US, boreal forests of Canada, northern Eurasia, and northeastern China.

While Figures 2b and 2c provide information pertaining to the sensitivity of Figure 2a to different threshold values, supplementary Figures S2 and S3 provide evidence that confining the analysis to smaller regions (i.e., extratropics and North America) does not substantively alter the results presented in Figure 2a. Generally, there are no significant differences in the spatial patterns of strong TLM values in the Northern Hemisphere during the strongly coupled seasons (MAM, JJA, and SON) when the analysis region is the entire globe (Supplementary Fig. S2c, S2d, and S2g) or is limited to just the Northern extratropics (Supplementary Fig. S3c, s3d, and S3g). Some differences emerge in DJF because L-A coupling is weak in Northern Hemisphere in winter. The quantile analysis at the global scale can help us to ignore those weakly coupled regions. All in all, Figures 2, S2, and S3 demonstrate that the key results based on the quantile analysis are not particularly sensitive to changes in the analysis region or the quantile threshold.

## 3.2 M vs E

Through analyzing the formulas of $TLM_E$ and $TLM_M$ (Sect. 2.4), we demonstrate that both the $\sigma$ term and the numerator of the $\rho$ term (denoted by $N(\rho)$) attenuate from $TLM_E$ to $TLM_M$. The decreasing rate relies on the contrast between the variation of daily elements within the same month and the variation of daily elements across months. Furthermore, we infer that the memory of specific E time series (i.e., $\overline{ACF}_{>75\%}$, see Sect. 2.4) or paired E time series (i.e., $\overline{CCF}_{>75\%}$ and $\overline{CCF}_{<25\%}$ for positively and negatively correlated pairs, respectively) can be an indicator characterizing the coupling signal loss from E to M.

Figure 3 verifies our deduction by showing statistically significant correlations between the coupling signal loss rate and the indicator regarding L-A memory. These significant correlation coefficients suggest that our indicator can capture the global pattern of coupling signal attenuation due to monthly smoothing. Specifically, regions with higher auto-correlation between individual days lead to a more minor loss of information when a daily time series is converted to a monthly time series. In the negative pair case (Fig. 3d), the indicator sensitivity to the signal attenuation may be weakened. The primary distractors (top and bottom-right regions isolated by blue lines in Fig. 3d) are from areas with extreme climate conditions, such as Greenland, Sahara, and Arabia (Fig. 3f). Nevertheless, the significance of the correlation coefficient suggests that the indicator is still able to reflect the attenuation magnitude. Surprisingly, the indicator captures not only the signal attenuation, but also phase shifts (the negative quadrant in Fig. 3e).

Through Figure 3, we demonstrate that $TLM_M$ loses L-A coupling signal as a result of smoothing the E time series and the memory of L-A states significantly affects the attenuation process. Although memory is another facet of coupling at the seasonal scale (Dirmeyer et al., 2009, 2016, 2018; Guo et al., 2011), it is not the main focus of TLM diagnosing the inter-daily L-A interactions. Moreover, two types of memory (auto-correlation of a single variable and cross-covariance of coupled variables) jointly influence the $TLM_M$ in the form of the quotient (Eq. 6 and 7), which increases the uncertainty of $TLM_M$ reflecting the signal of local L-A memory. Thus, the diagnoses based on $TLM_M$ are obscured by the varied memories of the L-A state, leading to a bias in the discovered hot spots of L-A coupling. Some regions with strong L-A coupling but low L-A memory (i.e., large daily fluctuations) may be overlooked by $TLM_M$.

## 3.3 E vs D

The value of $|\mathcal{L}_D|$ is larger than $|\mathcal{L}_E|$ worldwide (Supplementary Fig. S4a), and the primary contributor is the variability ($C_\sigma$, Fig. 4a). But the universal increase of $C_\sigma$ is not always the key driver of spatial pattern differences between $\mathcal{L}_E$ and $\mathcal{L}_D$ (Fig. 4c). For instance, both $\mathcal{L}_E$ and $\mathcal{L}_D$ suggest a portion of middle and high latitude regions of the Northern Hemisphere with strong soil moisture-sensible heat flux ($\theta-H$) coupling (Supplementary Fig. S5). However, different from $\mathcal{L}_E$, $\mathcal{L}_D$ suggests stronger coupling in North America than in Eurasia, which is primarily caused by the change of $\rho$ ($C_\rho$ and $C_{\sigma\rho}$). This difference is caused by the time averaging algorithm of the E time series, which considers one day from 0:00 to 24:00 based on Coordinated Universal Time (UTC). Thus, the E averaging period in the Western Hemisphere starts at night and ends on the following day. The opposite is true for the Eastern Hemisphere (left panel of Fig. 4e). However, in a large region of North America, the

nighttime soil moisture $\theta_N$ is more correlated to the daytime soil moisture $\theta_N$ of the previous day than the next day (Supplementary Fig. S6). Thus the entire-day average in the Western Hemisphere dramatically flattens the inter-daily fluctuations of soil moisture, leading to an underestimation of $\rho(\theta, H)$ by E. The right panel of Figure 4e shows that in a selected area of North America, the difference between E- and D- based $\rho(\theta, H)$ is significantly reduced if the $\theta_E$ was calculated by averaging the $\theta_D$ and the following $\theta_N$.

Figure 4b shows that both $C_\sigma$ and the $C_\rho$ can be important for $\Delta|\mathcal{A}|$ from E to D. $C_\sigma$ is likely the main contributor in humid regions, while the $C_\rho$ dominates arid and semi-arid areas. Figure 4d illustrates that $C_\sigma$ is the primary contributor to quantile increase in most strong $\mathcal{A}$ regions (yellow areas in Figure 2c). However, in fact, their quantile increase is caused by the quantile decrease in the Sahara and Arabia (Supplementary Fig. S4b), where $\mathcal{A}$ is negative (Supplementary the second row of Fig. S7). As $\mathcal{A}_D$ is universally higher than $\mathcal{A}_E$, the coupling strength over the Sahara and Arabia is weakened.

Generally, the land surface is the source of heating for the lower atmosphere during the day. Driven by the surface temperature $T_s$, $H$ heats the air and grows the height of the PBL (left panel of Fig. 4f), leading to positive $\rho(H, T_{2m})$ and $\rho(H, P_{lcl})$. However, the climate of the Sahara and Arabia is likely dominated by another mechanism. Over the northern Sahara, for instance, atmospheric advection seems to be the primary driver of inter-daily variations of near-surface atmospheric states (i.e., both $T_{2m}$ and $D_{2m}$) instead of the surface (middle panel of Fig. 4f, see Supplementary Text). A key consequence is that the $T_{2m}$ is no longer a passive variable, but drives the $H$ fluctuation (right panel of Fig. 4f), resulting in a negative $\rho(H, T_{2m})$ and further a negative $\rho(H, P_{lcl})$. In fact, both the bottom-up heating and the advection-driven heating mechanisms (left and middle panels of Fig. 4f) affect the climate variations in this region. However, the former only occurs during the daytime, while the latter can exist throughout a day. In comparison to E, the D averaging approach can minimize the effect of the former in L-A diagnoses.

## 4   Discussion

We demonstrate that the use of both monthly-mean and entire-day-mean daily data may result in biases in the diagnosis of L-A coupling. By comparing the two-legged metrics (TLM) calculated by the monthly (M), the daytime-only-mean (D), and entire-day-mean (E) time series, we found that the coverage discrepancy of their spatial patterns of strong coupling can be as large as 84.8% (Fig. 1). The diagnostic uncertainties introduced through monthly smoothing (i.e., differences between TLM$_E$ and TLM$_M$) are determined by the persistence or memory of local L-A states, which may result in the overestimation of L-A coupling strength in some climatic transition zones where climatic inter-monthly variations are larger than intra-monthly variations. Furthermore, we have demonstrated that integrating nighttime information in L-A diagnoses (i.e., TLM$_E$) may incorporate confounding effects from other mechanisms.

Although monthly-based and daily-based correlation coefficients capture the synchronized fluctuations of two variables from different perspectives, their linkage is yet unclear. In this study, for the first time as far as we know, we demonstrate mathematically how the correlation is weakened by monthly smoothing. Moreover, we propose indicators based on the auto-correlation function and cross-correlation function representing L-A memory to characterize the information loss. And these indicators are

able to capture the information loss worldwide regardless of geophysical and atmospheric complexities (Fig. 3). In addition, these indicators first link the memory of time series to the correlation attenuation due to coarser temporal smoothing, which has potential implications in broad fields.

Two mechanisms obscuring L-A diagnoses are discovered for the first time in our study, which again reflects the crucial need for daytime-only-mean data. First, atmospheric advection may dominate the daily fluctuations of both sensible heat flux and the LCL height in the Sahara and Arabia, resulting in a spurious negative relationship between the two. In comparison to highlighting these trivial regions by daily data-based diagnosis, daytime-only-mean data can make the diagnosis avoid this pitfall. Second, the traditional entire-day-mean daily data is obtained by averaging over 24 hours based on the UTC. It emphasizes shifted diurnal cycles according to longitude, which may mask signals of land state fluctuation in the Western Hemisphere, and provide inconsistent comparisons with the Eastern Hemisphere.

Land-atmosphere interactions have been demonstrated to be a key element in understanding climate dynamics (Berg et al., 2017; Findell et al., 2015; Humphrey et al., 2021; Koster et al., 2004; Seneviratne et al., 2010; Taylor et al., 2012). Different from simple causality, the land and the atmosphere are highly coupled by multiple variables that interact with each other (Santanello et al., 2018; Seneviratne et al., 2010), which raises difficulties for the understanding and simulation of relevant processes (Taylor et al., 2012, 2017). To investigate the complex coupled system, we must characterize its behaviors under various conditions and reveal relevant physical processes. Thus, a suite of metrics has been proposed to detect the features of a specific process (Santanello et al., 2018) based on either physical or statistical perspectives (https://www.pauldirmeyer.com/coupling-metrics). These metrics are helpful to evaluate model performance either against observations or through model inter-comparisons, and further support model improvements. However, it is rare to find datasets providing the required complete fields of high-frequency ($\leq 3$ hours) outputs for L-A investigations. For instance, daily data is generally the highest frequency output provided by numerous model inter-comparison projects (e.g., Eyring et al. (2016); Warszawski et al. (2014)), which is not adequate to diagnose the performance of Earth system models (ESMs) in simulating L-A interactions. Moreover, our study demonstrates that even daily data may overlook some important L-A patterns due to the perturbations of other processes.

Therefore, we call for careful attention to the requirements of high-frequency data in terms of diurnal cycle investigations, whose diagnoses can further reinforce ESM skills in predicting future climate under different scenarios. Assuredly, storage is a bottleneck for producing and sharing high-frequency data. Thus, we propose two approaches to balance the cost of storage and the need for high-frequency data. One approach is to integrate process-based metrics within ESMs so that the metric values themselves can be saved as model output, rather than calculated *a posteriori* (Findell and Eltahir, 2003a, b; Santanello et al., 2009; Tawfik and Dirmeyer, 2014). Therefore the diagnostic information can be easily collected at the cost of only a little extra computing time. The other is to generate different types of daily model output for different research purposes. In addition to daytime mean values, separate averages throughout the local morning, midday, afternoon, and nighttime would be interesting as well depending on the specific perspectives of interest (Taylor et al., 2012; Guillod et al., 2015). Such averaging algorithms must depend on the local time rather than the UTC, and the varied daytime length according to latitude and time of year should be considered.

# 5 Conclusions

This study demonstrates that the use of monthly or entire-day-mean daily data may lead to uncertainties in diagnoses of land-atmosphere (L-A) coupling strength and interactions. The arithmetic mean of time series including the nighttime weakens the signal of L-A coupling. And the spatial heterogeneity of such weakening effects can alter the diagnosis of coupling strength based on the two-legged metrics. In addition, two phenomena were discovered, which can dramatically obscure the L-A diagnoses if the entire-day-mean daily time series is applied. One is a spurious relationship between flux and atmosphere states led by atmospheric advection in Sahara and Arabia. The other is the underestimation of L-A coupling in the Western Hemisphere due to the classical daily averaging algorithm based on the Coordinated Universal Time that twists the segmentation of the diurnal cycle. Through this study, we call attention to the requirements of high-frequency data for L-A diagnoses. L-A metrics can be either integrated within Earth System Models to avoid huge storage for high-frequency outputs or fed by outputs averaging over the sub-daily period of interest. Either of the approaches can improve the accuracy of L-A diagnoses with minimal cost of computing time and storage space.

*Code and data availability.* The $0.5°$ ERA5 data is available at https://cds.climate.copernicus.eu/#!/home. The code for calculating the two-legged metrics can be found via http://www.coupling-metrics.com.

*Author contributions.* Z.Y., K.F., E.S., and S.M. conceived the research question; Z.Y. and K.F. designed the research; Z.Y. and N.R. collected and processed the data; Z.Y. performed the research; Z.Y. drafted the manuscript; all authors interpreted results and contributed to the editing.

*Competing interests.* The authors declare no competing interests.

*Acknowledgements.* Z.Y. and K.G. are supported by CIMES grant NA18OAR4320123. P.D. is supported by NOAA grant NA19OAR4310242. Yujin Zeng and Jing Feng are acknowledged for their comments and suggestions on an internal review of the manuscript. We acknowledge GFDL resources made available for this research. We thank the European Centre for Medium-Range Weather Forecasts (ECMWF) for providing the ERA5 data. We thank the editor and two anonymous referees for their helpful comments and efforts.

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

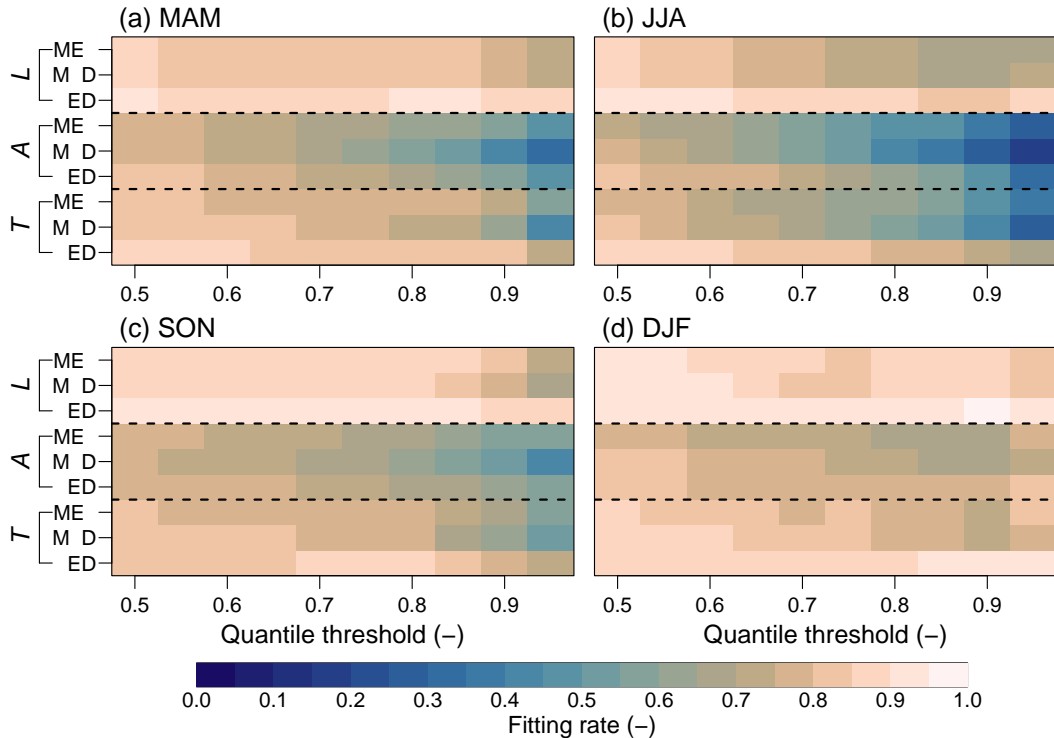

**Figure 1.** Fitting rates of different paired comparisons as a function of quantile threshold by using global data (see Sect. 2.3). The subplots represent different seasons. The three bands (separated by dashed lines) in each subplot indicate the land leg ($\mathcal{L}$), the atmospheric leg ($\mathcal{A}$), and the total ($\mathcal{T}$). Within each band; the three rows represent three paired comparisons, they are (from top to bottom) M vs E, M vs D, and E vs D.

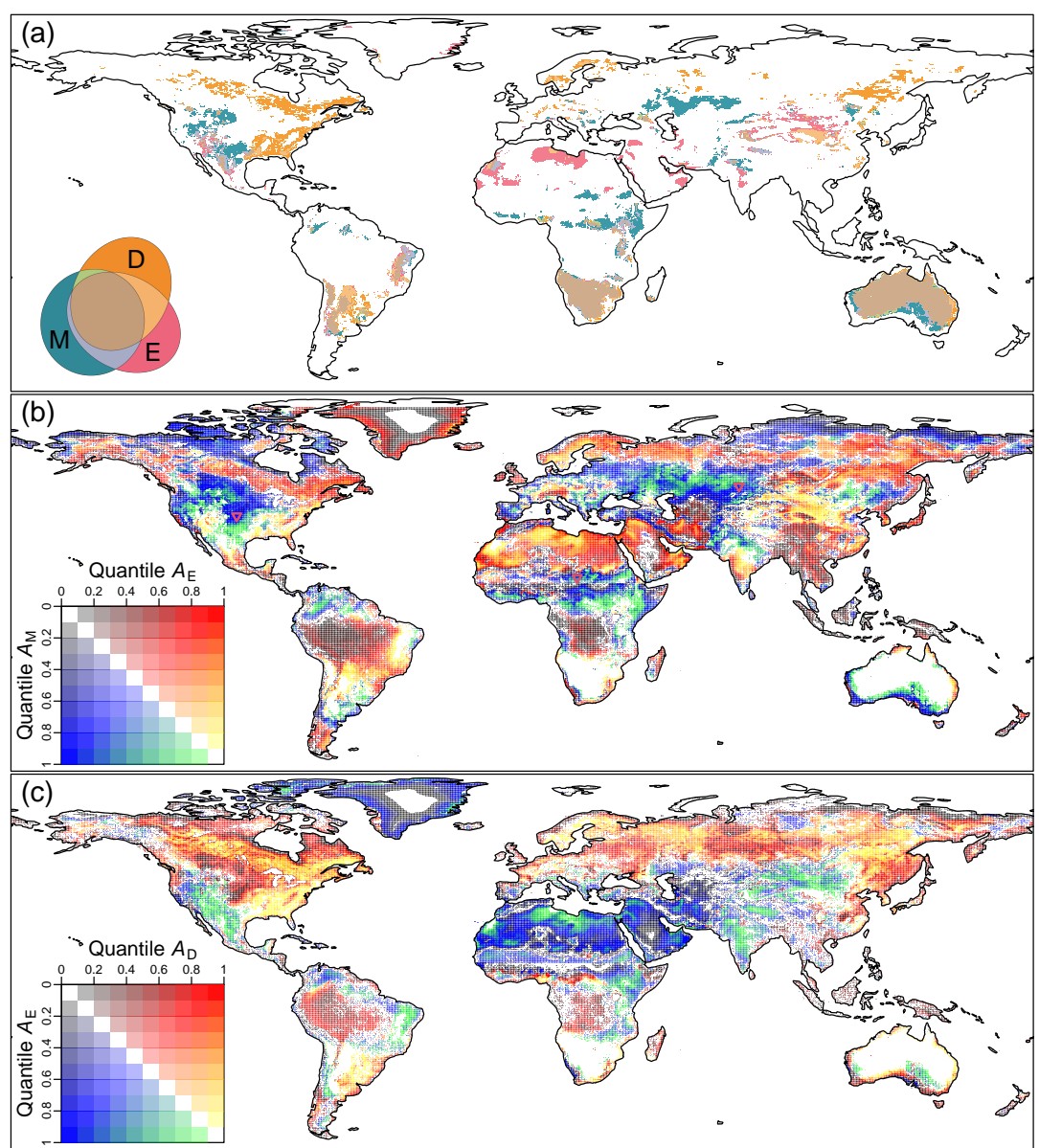

**Figure 2.** (a) Spatial patterns of significant $\mathcal{A}_M$, $\mathcal{A}_E$, and $\mathcal{A}_D$ (top $10\%$ quantile of absolute values) in summer (JJA and DJF for northern and southern hemisphere respectively). Euler diagrams show the colors for specific relationships (intersections, unions, or disjoints) among $\mathcal{A}_M$, $\mathcal{A}_E$, and $\mathcal{A}_D$, and the areas of colored patterns also correspond to the fractions. (b)-(c) Quantile changes (b) from $\mathcal{A}_M$ to $\mathcal{A}_E$ and (c) from $\mathcal{A}_E$ to $\mathcal{A}_D$ in summer. The quantile of the $\mathcal{A}$ is separated into ten bins. The color of the grid cell is explained by the legend, where $x$- and $y$-axes indicate its quantile bins of specific $\mathcal{A}$. The diagram has three aspects of information. First, warm (cold) colors indicate quantile increase (decrease) from the original $\mathcal{A}$ ($y$-axis) to the final $\mathcal{A}$ ($x$-axis). Second, the smaller the quantile difference is, the more transparent the color. White indicates no change of quantile bin. Third, as the shifts in the large quantile bins are the main focus, we highlight this part in green and yellow. For shifts that occur within the low quantile bins, colors fade to gray. Three red triangles are samples from three regions where $\mathcal{A}$ is dramatically underestimated by monthly smoothing.

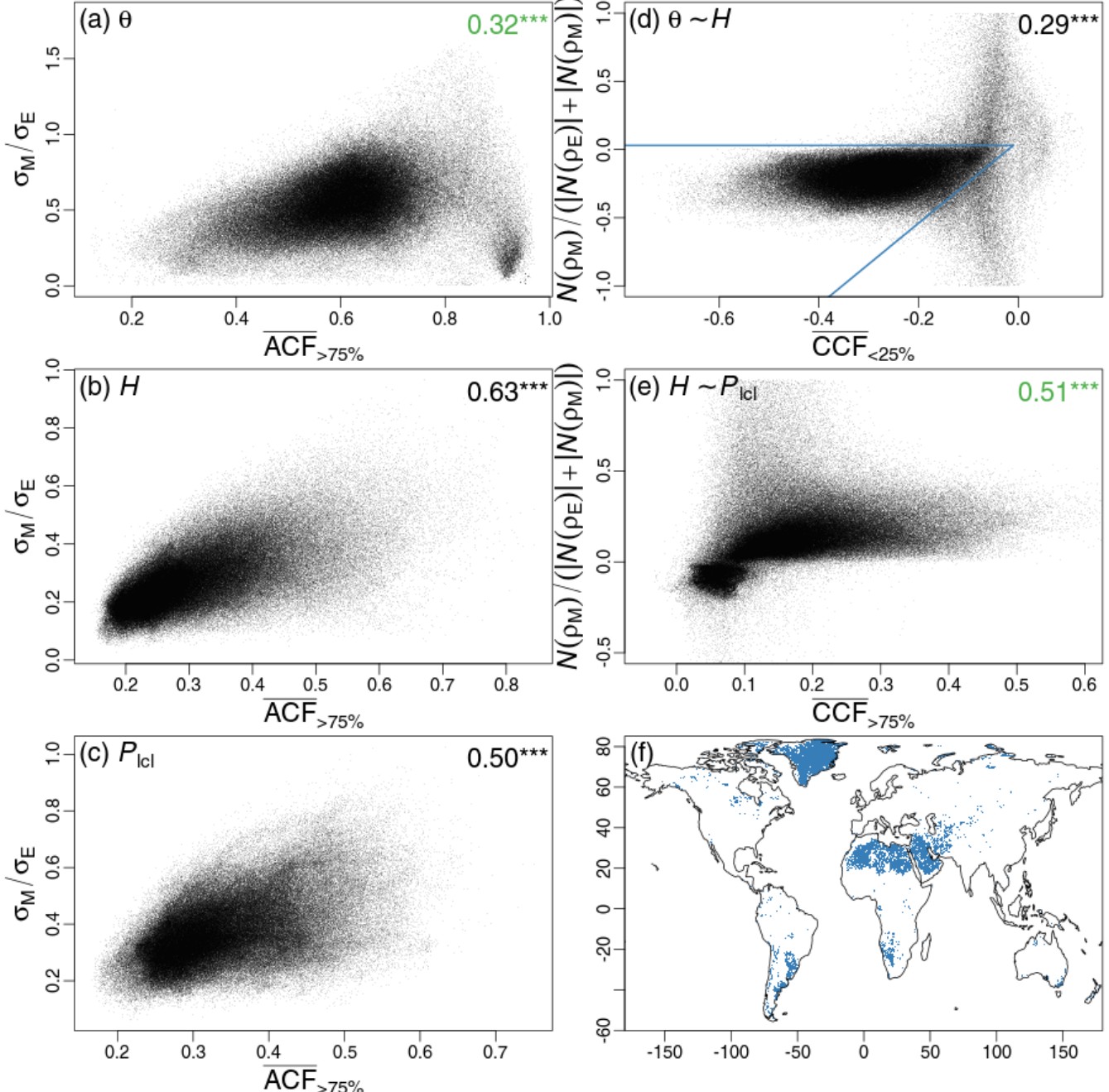

**Figure 3.** Scatter plot of coupling signal loss rate when moving from $\text{TLM}_E$ to $\text{TLM}_M$ as a function of an indicator reflecting the memory of L-A states. Points represent terrestrial grid cells around the globe. (a)–(c) Loss rate of the $\sigma$ term as a function of averaged auto-correlation function ($\overline{\text{ACF}}$) with quantile larger than 75% (see Sect. 2.4). (d)–(e) Loss rate of the numerator of the $\rho$ term (see Sect. 2.4) as a function of averaged cross-covariance function ($\overline{\text{CCF}}$) within a certain quantile range (shown by the subscript, see Sect. 2.4). Dark and green values at the top right are Person and Spearman correlation coefficients for linear and nonlinear relationships, respectively. $^{***}$ indicates $p < 0.001$. (f) Patterns with values out of the main cluster (separated by two blue lines) in (e).

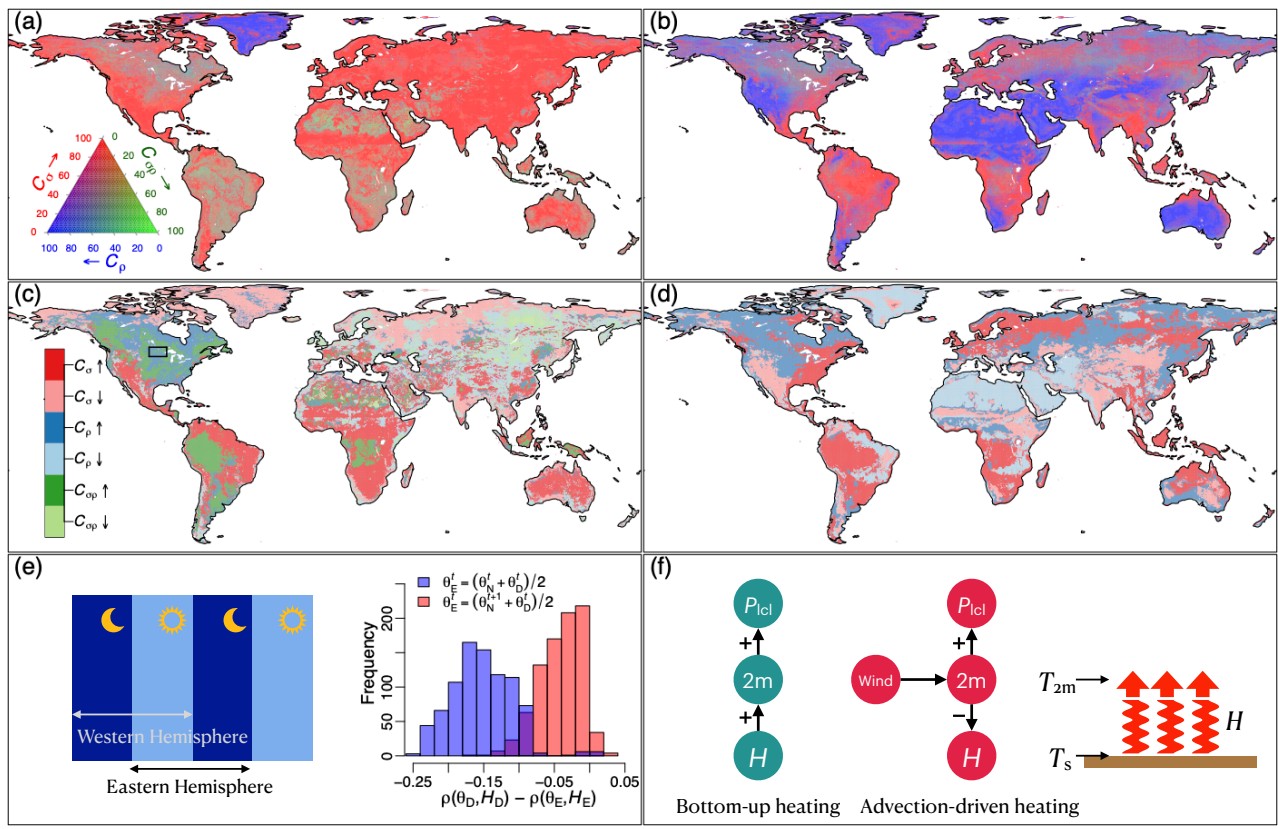

**Figure 4.** Comparison between TLM$_D$ and TLM$_E$. Left panel: the land leg ($\mathcal{L}$); right pane: the atmospheric leg ($\mathcal{A}$). Top row: fractions of the three components of $\Delta|M|$ ($|M_D| - |M_E|$, Eqs. 8 and 9, see Sect. 2.5). Red, blue, and green indicate contributions of fluctuation, correlation, and joint of the two ($|C_\sigma|$, $|C_\rho|$, and $|C_{\sigma\rho}|$), respectively (see Sect. 2.5). Middle row: primary contributor to pattern shift in TLM (see Sect. 2.6). The legend contains three pairs of colors: red, blue, and green indicate $C_\sigma$, $C_\rho$, and $C_{\sigma\rho}$ as the primary contributor, respectively. A darker (lighter) color indicates a quantile increase (decrease) from E to D. Left panel of (e): conceptual figure showing the combinations of daytime and nighttime that make up the E time series in the Eastern versus Western Hemisphere. Right panel of (e): histograms of the difference between D- and E-based $\rho(\theta, H)$. Data is from the rectangle region shown in (c). The blue histogram indicates the cases with the original $\theta_E$ (an average of the nighttime soil moisture $\theta_N$ and the following daytime soil moisture $\theta_D$). Red histogram indicates the cases with the modified $\theta_E$ (an average of the $\theta_D$ and the following $\theta_N$). Left and middle panel of (f): two mechanisms driving the $\mathcal{A}$. Right panel of (f): the definition of sensible heat flux $H$ which reflects the temperature gradient from the surface to the near-surface (2m).