# Peer review of "Daytime-only-mean data enhances understanding of land-atmosphere coupling"

_EGUsphere, 2022_

## Author Comment (AC1)

**Reply to Referee #1 comments on "Daytime-only-mean data can enhance understanding of land-atmosphere coupling"**

Zun Yin on behalf of co-authors

**The paper tackle an important and yet unexplored dependency of land-atmosphere coupling metrics from the choice of daily monthly or daytime only subsets.**

**The paper is well written and reaches a number of conclusions that are relevant for model diagnostics. In particular the use of daytime only time-series can provide a more accurate detection of regions of strong coupling.**

AR: Thank you so much for these positive comments. Please check our reply below. Referee's comments are in bold; authors' responses are in regular; and modifications in the manuscript are in blue.

**L270: add the two key discoveries in the conclusions.**

AR: Thanks for reminding us. In the revision, we will summarize these two newly-discovered mechanisms in the conclusion, as:

...In addition, two phenomena were discovered, which can dramatically obscure the L-A diagnoses if the entire-day-mean daily time series is applied. One is a spurious relationship between flux and atmosphere states led by atmospheric advection in Sahara and Arabia. The other is the underestimation of L-A coupling in the Western Hemisphere due to the classical daily averaging algorithm based on the Coordinated Universal Time that twists the segmentation of the diurnal cycle...

---

## Author Comment (AC2)

**Reply to Referee #2 comments on "Daytime-only-mean data can enhance understanding of land-atmosphere coupling"**

Zun Yin on behalf of co-authors

**This paper demonstrates that the averaging approach employed by models (in this case, ERA5) in generating output diagnostics has an impact on what can be inferred from those diagnostics in the context of land-atmosphere coupling strength. Intuitively, such an impact makes perfect sense. I agree with the authors' call to modeling centers to provide relevant land-atmosphere coupling diagnostics at higher time resolution for improved analysis of land-atmosphere coupling.**

AR: Thank you so much for your review and comments. We very much appreciate your agreement with our vision. Please check our point-by-point replies below. Referee's comments are in bold; authors' responses are in regular; and modifications in the manuscript are in blue.

**All this being said, I must recommend major revision for this paper. The analysis strategy used is far from intuitive, and after reading the paper several times, I'm left unconvinced that the particular strategy used here is optimal (though I don't pretend to know what the optimal strategy is). It almost goes without saying that daytime-only data can get at the two-legged metric better than full-day or full-month data; still, I can't wrap my head around the idea that the quantile approach is the best way to tell us what we want to know (see comment 2 below).**

AR: The quantile analysis is selected because of two features of the two-legged metrics (TLM): (1) The coupling strength estimated by TLM is reflected by the relative spatial difference rather than the numerical values of TLM, because there are not fundamental threshold values that distinguish regions of strong coupling from regions of week coupling. The primary aim of TLM is to discover hot spots where land and atmosphere are strongly coupled over the globe. Therefore, the spatial pattern (i.e., relative difference) of TLM is more important than the sole values of TLM in grid cells. (2) The magnitudes of a TLM, dominated by the standard deviation term (i.e., $\rho$ in Eq. 1), based on different timing periods are essential different. For instance, the daily amplitude of entire-day-mean sensible heat flux is systematically smaller than that of daytime-only-mean sensible heat flux. Therefore, the TLMs based on the raw values of daytime-mean (D), entire-day-mean (E), and monthly-mean (M) cannot be directly compared to demonstrate how the spatial patterns shift from one diagnosis to another.

The quantile approach can reflect the spatial patterns of TLM and provide the possibility of patterns comparison between TLMs based on different time smoothing data. In fact, other climate-relevant studies have successfully utilized the quantile approach to compare estimates based on different algorithms. For example, because satellite-based and modeled estimations are not suitable to compare to gauge measurements directly, the quantile approach was employed for relevant bias correction

or downscaling in the form of a probability density function (PDF) (Guo et al., 2018; Vrac et al., 2012; Xie et al., 2017). More importantly, the spatial differences highlighted by quantile analysis prompted the discovery of two physical mechanisms obscuring L-A diagnoses in this study (Sect. 3.3 and Fig. 4 in the main text). These findings prove that the quantile analysis is useful for spatial difference investigation in our case.

As the main concern of the L-A coupling is the daytime period, we assume that the D data are more reliable than the E and the M. Again, we found two mechanisms disturbing the L-A coupling diagnoses. However, these disturbances can be dramatically reduced if daytime-only data are utilized, confirming that the D is more reliable for L-A diagnoses than the E.

In the revision, we will explicitly explain the features of TLM and the reason for applying the quantile approach, as:

The TLMs are designed to highlight differences in L-A coupling strength between geographic regions and/or between different times of year in a given region. Those relative differences require subjective decisions to determine the threshold values separating regions of "strong" coupling from regions of weaker coupling. However, direct comparison of the numerical values of TLMs based on different time windows of inputs (i.e., M, E, and D) is not appropriate for three primary reasons. First, the magnitude of the TLMs is strongly affected by the $\sigma$ term (Eq. 1), and this measure of variability can be quite different for daytime and nighttime processes. For example, $H_D$ has much larger variance than the $H_E$, which systematically enlarges the $\mathcal{A}_D$. Additionally, strong L-A coupling signals can be positive or negative, suggesting that the change of TLM's magnitude (its absolute value) is the relevant quantity of interest rather than the magnitude of changes. Finally, L-A coupling processes are not characterized by clear thresholds, but rather by relative spatial and temporal differences.

To overcome these limitations and remove any subjectivity in our assessment of coupling strength, we use quantile to assess coupling strengths and quantify the spatial differences between $TLM_M$, $TLM_E$, and $TLM_D$.

**1. Because the quantile analysis approach is not intuitive, further exposition in the Methods section would go a long way toward making this study more comprehensible. Perhaps the authors have spent so much time thinking about the analysis approach that it comes as second nature to them, but they should know that this won't be the case for the average reader. Significant additional explanation is needed. For example, I'm guessing that quantiles are based on all land (non land-ice) points across the globe. True? Please clarify. Also, are the quantiles computed separately for each season? If so, why are southern hemisphere JJA points mixed in with northern hemisphere JJA points in determining the quantiles? One would think that seasonal variations in the diagnostics would be hemisphere-specific.**

AR: The quantile analysis was applied over land defined by the ERA5 data (e.g., land-sea fraction larger than 0.5), so land-ice is included. Moreover, the quantile analysis was firstly applied based on seasons (Fig. 1). After demonstrating the largest discrepancies existing in summer, we conducted the following analysis focusing on summer only (JJA and DJF for Northern and Southern Hemisphere, respectively). We will clarify these points in the text as:

We collected ERA5 output over land (land-ice included) every other hour from 1:00 UTC...

To focus on the season and coupling leg with the largest sensitivity to time series averaging window, we select $\mathcal{A}$ in summer (JJA and DJF in the Northern and Southern Hemisphere, respectively) as an example to explore the TLM differences in the

following content. Figure 2: (a) Spatial patterns of significant $\mathcal{A}_M$, $\mathcal{A}_E$, and $\mathcal{A}_D$ (top 10% quantile of absolute values) in summer (JJA and DJF for northern and southern hemisphere respectively).

**2. Assuming that I do know what the authors are doing, I have some misgivings about what the quantile approach can tell us. Would consideration of only northern hemisphere extratropical points (probably a much cleaner approach, given seasonality) give the same results? Would a continental-scale analysis (e.g., North America only) give the same results? There's no way of knowing a priori; one can only speculate. Also, consider two highly hypothetical scenarios:**

AR: We compared the quantile analysis over specific regions (i.e., the extratropical region in Northern Hemisphere and the North America) and over the globe in Figure R1, R2, R3, and R4. Generally, there is no significant differences in terms of spatial patterns in L-A strongly couple seasons. The patterns fade in DJF in these global analysis because the L-A coupling is weak in Northern Hemisphere in winter. And the quantile analysis at global scale can help us to ignore those L-A weakly coupling regions, which is more advanced than the regional analysis. All in all, the key results based on the quantile analysis will not be very sensitive to changes in the analysis region or the quantile threshold.

**a) The TLM values produced with all three averaging approaches are perfectly valid (i.e., are perfectly consistent with each other) except over 20% of the Earth (defined by vegetation type, location on the globe, or whatever). In that 20%, the monthly averaging approach inappropriately assigns a very high coupling strength when the actual coupling strength is very low. In this hypothetical example, the monthly averaging approach would look very bad at the high extreme, as it should, but it would also look bad (20% off) everywhere else, when this example's assumptions say that it actually works just fine. This seems to be a basic limitation of the quantile approach.**

AR: The key aim of this study and the quantile analysis is to demonstrate whether diagnoses based on different time smoothing data can provide the same spatial patterns. As mentioned in the hypothesis, spatial difference will be found between M- and D-based (for example) diagnoses by quantile analysis because monthly data fails reflect the high extreme (the top 20% in reality is not fully equal to the top 20% in M-based TLM). Therefore, our aim has been reached. Specifically, through showing spatial differences, we demonstrate that at least one of the diagnosis contains bias. According to our assumption (D-based diagnosis is more accurate than E- and M-based diagnosis), we can conclude that the monthly averaging approach inappropriately assigns coupling strength somewhere.

**b) In a separate hypothetical example, suppose that 80% of the globe experiences no land-atmosphere feedback of any relevance at all. In this case, quantile differences found between the averaging approaches within this lower 80% would have no practical meaning, and there'd be no point, e.g., in plotting quantile changes.**

AR: First, we demonstrated that L-A coupling existed in a broad regions where the $\rho$ term in TLMs is significant (colored regions in Fig. S4), unless the referee believes that a significant correlation does not mean coupling at all. Even if the assumption holds, we provided spatial difference analysis (the fitting rate) based on different quantile thresholds (from 50% to 95% Fig. 1). As there is no specific threshold to determine whether L-A feedback exists or not, we let the readers interpret the result by selecting the threshold by themselves. Nevertheless, we think the top 10% TLMs indicates that the coupling reaches a certain

strength, which was used as the threshold in Figure 2a.

**I'm not saying that these scenarios are realistic; I'm just saying that it's easy to come up with scenarios that call into question the understanding that can be gained from a quantile-based analysis. The authors should provide significant discussion about the limitations of dealing with quantiles like this.**

AR:In fact, the quantile analysis can reach our aim in both scenarios proposed above. True. Many scenarios can be proposed to debate that the quantile analysis may over/underestimate coupling strength at a certain level. However, the key aim of the quantile analysis is to demonstrate whether spatial differences exist between different diagnoses. Thus we think the quantile analysis is suitable for our study unless a case showing that two diagnoses have the same spatial patterns but result in spatial pattern differences in quantile analysis.

**3. I disagree with the conclusion on lines 234-236, in reference to the Koster et al. study. That study did not use the two-legged approach to quantify coupling; it simply quantified the impact of soil moisture variations on precipitation variability at the multi-day time-scale. For the particular coupling characterization it was after, the calculation was exact and was not limited in any way by daytime-only vs. all-day vs. multi-day considerations. The results of the present study are best considered in relation to studies that use the two-legged metric.**

AR: We agree with the referee that TLM and Koster's approach provide L-A interaction diagnoses from different angles and different temporal scales. We will modify the manuscript accordingly to focus on a broad topic regarding monthly smoothing, as:

..., which may result in the overestimation of L-A coupling strength in some climatic transition zones where climatic inter-monthly variations are larger than intra-monthly variations.

**4. Section 2.4 came off as opaque to me. What does the "top 25% quantile" refer to - if it refers to the ACF values, why are the lower values being ignored? Why is the ratio of the sigmas relevant? What is meant by "numerator of the rho term"? Why is the relevance of the ratio of the N terms? Also, though I can kind of guess what are the authors getting at when they talk about signal attenuation in the first place, I can't be sure. A major rewrite is needed here.**

AR: We apologize for failing to include relevant details in the main text. The relevant information was available in the supplemental materials, but we fully agree with the author that the main text should be interpretable without needing to refer to the supplement. We will move this material to the main text in the revision. Moreover, the $\sigma_M/\sigma_E$ does not indicate attenuation rate but the rate of information maintenance. In the revised manuscript, "attenuation resistance" is used to replace "attenuation rate".

In Sect. 2.4, we will add,

[revised manuscript text omitted]

**5. The correlations in Figure 3 are undoubtedly statistically significant, but they are far from "high" (line 185) or even "moderately large" (line 191). Those in panels (a) and (b) indicate only a 10% explanation of variance, and those in the remaining panels indicate well less than half the variance explained. The text, though, presents these fields as clear indications that the authors have identified the main controls on various quantities ("Significant correlation coefficients suggest that our indicator adequately explains the attenuation..."). To be honest, I got very little out of Figure 3 and the associated discussion.**

AR: We agree that some adjectives such as "high", "moderately large" are subjective. In the revised manuscript, we will use "significant" instead. Through Figure 3 we, for the first time as far as we know, demonstrated why information is missing through monthly smoothing and how to characterize the degree of missing by an understandable concept. As far as we knew, what the difference between daily correlation and monthly correlation represent has not been clearly answered yet. Through analyzing their formulas (see our reply to the previous comment), we demonstrate how the correlation information is weakened by monthly smoothing mathematically. Based on this analysis, we propose the indicator $\overline{\mathrm{ACF_\%}}$ representing L-A memory to characterize the information loss. Although the indicator is not perfect, it is the best way to compress memory information (an array of correlation coefficients) into one value currently. More importantly, Figure 3 shows that the indicator is able to capture the information loss worldwide regardless of geophysical and atmospheric complexities. In addition, this indicator firstly links the time series memory to the correlation attenuation due to coarser temporal smoothing, which has potential implications in broad fields.

In the revision, the novel indicator and its advantages will be added in the discussion, as:
Although monthly-based and daily-based correlation coefficients capture the synchronized fluctuations of two variables from different perspectives, their linkage is yet unclear. In this study, for the first time as far as we know, we demonstrate how the correlation is weakened by monthly smoothing mathematically. Moreover, we propose indicators based on the auto-correlation

function and cross-correlation function representing L-A memory to characterize the information loss. And these indicators are able to capture the information loss worldwide regardless of geophysical and atmospheric complexities (Fig. 3). In addition, these indicators first link the memory of time series to the correlation attenuation due to coarser temporal smoothing, which has potential implications in broad fields.

**6. Would it be appropriate to at least mention that the daytime-only diagnostics may produce different results from midday-only diagnostics (e.g., 10AM-2PM)? Presumably not much coupling occurs at dusk and dawn. I'm not suggesting that midday diagnostics be examined in this paper; it's just that the overall problem of optimal averaging time goes beyond simply comparing all-day diagnostics to daytime-only diagnostics.**

AR: Thanks for the proposal. We agree that the diagnoses based on different sub-time periods of the daytime may have significant differences. However, in comparison to the vast differences between the daytime and the nighttime, the diagnostic difference induced by different sub-time periods of the daytime is not the primary question, but is worth exploring in the following studies. Moreover, it is difficult to assume that midday-only data can further improve the diagnostics, because the key period of interest may vary with the specific process. For instance, the soil moisture in early morning may be coupled with the convective precipitation in the afternoon or early evening. Therefore, the optimal averaging time should be carefully investigated according to the process of interest.

In the revision, we will modify the text, as:

In addition to daytime mean values, separate averages throughout the local morning, midday, afternoon, and nighttime would be interesting as well depending on the specific perspectives of interest (Taylor et al., 2012; Guillod et al., 2015).

[Figure]

**Figure R1.** Spatial patterns of significant $\mathcal{L}_M$, $\mathcal{L}_E$, and $\mathcal{L}_D$ (top 10% quantile of absolute values) of different seasons in the extratropic region of the Northern Hemisphere. Euler diagrams show the colors for specific relationships (intersections, unions, or disjoints) among $\mathcal{L}_M$, $\mathcal{L}_E$, and $\mathcal{L}_D$. (a), (b), (e), and (f) are screenshots from the global quantile analysis. (c), (d), (g), and (h) are based on quantile analysis of the illustrated region.

[Figure]

**Figure R2.** Spatial patterns of significant $\mathcal{A}_M$, $\mathcal{A}_E$, and $\mathcal{A}_D$ (top $10\%$ quantile of absolute values) of different seasons in the extratropic region of the Northern Hemisphere. Euler diagrams show the colors for specific relationships (intersections, unions, or disjoints) among $\mathcal{A}_M$, $\mathcal{A}_E$, and $\mathcal{A}_D$. (a), (b), (e), and (f) are screenshots from the global quantile analysis. (c), (d), (g), and (h) are based on quantile analysis of the illustrated region.

[Figure]

**Figure R3.** Spatial patterns of significant $\mathcal{L}_M$, $\mathcal{L}_E$, and $\mathcal{L}_D$ (top 10% quantile of absolute values) of different seasons in the North America. Euler diagrams show the colors for specific relationships (intersections, unions, or disjoints) among $\mathcal{L}_M$, $\mathcal{L}_E$, and $\mathcal{L}_D$. (a), (b), (e), and (f) are screenshots from the global quantile analysis. (c), (d), (g), and (h) are based on quantile analysis of the illustrated region.

[Figure]

**Figure R4.** Spatial patterns of significant $\mathcal{A}_M$, $\mathcal{A}_E$, and $\mathcal{A}_D$ (top 10% quantile of absolute values) of different seasons in the North America. Euler diagrams show the colors for specific relationships (intersections, unions, or disjoints) among $\mathcal{A}_M$, $\mathcal{A}_E$, and $\mathcal{A}_D$. (a), (b), (e), and (f) are screenshots from the global quantile analysis. (c), (d), (g), and (h) are based on quantile analysis of the illustrated region.

---

## Author Response (AR2)

**Reply to Referee #2 comments on "Daytime-only-mean data can enhance understanding of land-atmosphere coupling"**

Zun Yin on behalf of co-authors

**The authors have improved their manuscript and have mostly addressed my earlier criticisms; I do not need to see the manuscript again. This said, I think they should do a little bit more to address the potential limitations of the quantile approach. (See my previous review.) Much of their response-to-reviewers was written to convince me of why their approach is valid without actually transferring equivalent arguments to the text. The paragraph that is added ("The TLMs are designed to highlight...") provides a justification for the approach but doesn't really address any pitfalls. As a reviewer, I represent the general reader who will likely still have the same concerns. As the text is currently written, for example, the general reader will still be wondering if a more regionally limited quantile analysis would give different results.**

AR: Thank you so much for taking the time to do the second round review. And we are glad to know that our responses have addressed your major concerns. We apologize for overlooking several key contents of our previous responses in the revision. In the new revision, apart from several minor revisions of the contents, we mainly improved the introduction and the discussion regarding quantile analysis. Via testing the quantile analysis at different regional scales, we demonstrated why the results in this study can avoid potential pitfall due to quantile analysis.

In the methodology, we explicitly explain why the quantile analysis, which has been widely used in different fields, is a suitable tool for the specific situation, as (Line 100 to 105) "The quantile approach can reflect the spatial patterns of TLM and provide the possibility of pattern comparison between TLMs based on different inputs. Other climate-relevant studies have also successfully utilized the quantile approach to compare estimates based on different algorithms. For example, because satellite-based and modeled estimations are not suitable for direct comparison with gauge measurements, the quantile approach was employed for relevant bias correction or downscaling in the form of probability density functions (PDF) Guo et al. (2018); Vrac et al. (2012); Xie et al. (2017)."

In the result section, we compare regional and global TLM diagnostics to demonstrate the consistency of the results based on the quantile analysis, as (Line 220 to 228) "While Figures 2b and 2c provide information pertaining to the sensitivity of Figure 2a to different threshold values, supplementary Figures S2 and S3 provide evidence that confining the analysis to smaller regions (i.e., extratropics and North America) does not substantively alter the results presented in Figure 2a. Generally, there are no significant differences in the spatial patterns of strong TLM values in the Northern Hemisphere during the strongly coupled seasons (MAM, JJA, and SON) when the analysis region is the entire globe (Supplementary Fig. S2c, S2d, and S2g) or is limited to just the Northern extratropics (Supplementary Fig. S3c, s3d, and S3g). Some differences emerge in DJF because L-A coupling is weak in Northern Hemisphere in winter. The quantile analysis at the global scale can help us to ignore those

weakly coupled regions. All in all, Figures 2, S2, and S3 demonstrate that the key results based on the quantile analysis are not particularly sensitive to changes in the analysis region or the quantile threshold." The attached two figures has been added to the online supplementary materials as Figure S2 and S3.

**References**

Guo, L.-Y., Gao, Q., Jiang, Z.-H., and Li, L.: Bias correction and projection of surface air temperature in LMDZ multiple simulation over central and eastern China, Advances in Climate Change Research, 9, 81–92, https://doi.org/https://doi.org/10.1016/j.accre.2018.02.003, including special topic on China Energy Modeling Forum, 2018.

Vrac, M., Drobinski, P., Merlo, A., Herrmann, M., Lavaysse, C., Li, L., and Somot, S.: Dynamical and statistical downscaling of the French Mediterranean climate: uncertainty assessment, Natural Hazards and Earth System Sciences, 12, 2769–2784, https://doi.org/10.5194/nhess-12-2769-2012, 2012.

Xie, P., Joyce, R., Wu, S., Yoo, S.-H., Yarosh, Y., Sun, F., and Lin, R.: Reprocessed, Bias-Corrected CMORPH Global High-Resolution Precipitation Estimates from 1998, Journal of Hydrometeorology, 18, 1617 – 1641, https://doi.org/10.1175/JHM-D-16-0168.1, 2017.

[Figure]

**Figure R1.** Spatial patterns of significant $\mathcal{A}_M$, $\mathcal{A}_E$, and $\mathcal{A}_D$ (top $10\%$ quantile of absolute values) of different seasons in the extratropic region of the Northern Hemisphere. Euler diagrams show the colors for specific relationships (intersections, unions, or disjoints) among $\mathcal{A}_M$, $\mathcal{A}_E$, and $\mathcal{A}_D$. (a), (b), (e), and (f) are screenshots from the global quantile analysis. (c), (d), (g), and (h) are based on quantile analysis of the illustrated region.

[Figure]

**Figure R2.** Spatial patterns of significant $\mathcal{A}_M$, $\mathcal{A}_E$, and $\mathcal{A}_D$ (top 10% quantile of absolute values) of different seasons in the North America. Euler diagrams show the colors for specific relationships (intersections, unions, or disjoints) among $\mathcal{A}_M$, $\mathcal{A}_E$, and $\mathcal{A}_D$. (a), (b), (e), and (f) are screenshots from the global quantile analysis. (c), (d), (g), and (h) are based on quantile analysis of the illustrated region.